# SmartSkeMa: Scalable Documentation for Community and Customary Land Tenure



**Malumbo C. Chipofya** [1,*] **, Sahib Jan** [2] **and Angela Schwering** [2]

1   Faculty of Geo-Information Science and Earth Observation (ITC), University of Twente, Hengelosestraat 99, 7514 AE Enschede, The Netherlands

2   Institute for Geoinformatics, Westfälische Wilhelms-Universität Münster, Heisenbergstrasse 2, 48149 Münster, Germany; s_jan001@uni-muenster.de (S.J.); schwering@uni-muenster.de (A.S.)

*   Correspondence: m.c.chipofya@utwente.nl; Tel.: +31-53-489-4760

**Abstract:** According to the online database landmarkmap, up to an estimated 50% or more of the world's habitable land is held by indigenous peoples and communities. While legal and procedural provisions are being made for bureaucratically managing the many different types of tenure relations in this domain, there continues to be a lack of tools and expertise needed to quickly and accurately document customary and indigenous land rights. Software and hardware tools that have been designed for documenting land tenure through communities continue to assume a parcel-based model of land as well as categories of land relations (RRR) largely dimensionally similar to statutory land rights categories. The SmartSkeMa approach to land tenure documentation combines sketching by hand with aerial imagery and an ontology-based model of local rules regulating land tenure relations to produce a system specifically designed to allow accurate documentation of land tenure from a local perspective. In addition, the SmartSkeMa adaptor which is an OWL-DL based set of rules for translating local land related concepts to the LADM concepts provides a more high-level view of the data collected (i.e., what does this concept relate to within the national LADM profile?) In this paper we present the core functionalities of SmartSkeMa using examples from Kenya and Ethiopia. Based on an expert survey and focus groups held in Kenya, we also analyze how the approach fairs on the Fit-for-Purpose Land Administration tools scale. The results indicate that the approach could be beneficial in scaling up mapping of community and customary lands as well as help reduce conflict through its participatory nature.

**Keywords:** customary land tenure; participatory mapping; fit-for-purpose land administration; land recordation tools; semantic technologies; land information system

## 1. Introduction

Much of the world's habitable land is held by indigenous peoples and communities. Surprisingly, there is a dearth of tools and expertise needed to quickly and accurately document customary and indigenous land rights. By contrast a lot of investment is put into implementation and improvement of statutory land administration systems as witnessed by many recent projects in the domain (LIFT in Ethiopia and the Rwanda LTR for example). This is ironic considering landmarkmap's estimate that of the nearly 70% undocumented tenures on land in the world a significant proportion is likely to cover up to 50% of the world's habitable areas (http://landmarkmap.org).

One possible explanation for the skewed investment towards securing state sanctioned forms of land rights is what James Scott referred to as the process of legibility making [1] (pp. 33–36). Legal land rights categories are often created to simplify (and make uniform) the relations of tenure on land to render them amenable to administration. This simplification inevitably entails a tearing down of some existing local social norms and cultural practices. As both Scott [1] and Abubakari et al. [2] note, this legibility making

often meets some forms of resistance. State systems can therefore co-exist or compete with existing norms, and may come into direct conflict with them.

Considering the impact of good land governance and tenure security on many of today's pressing problems (including environmental sustainability, food security, and economic and social development—the global Sustainable Development Goals (SDGs) acknowledge the impact of tenure security on all these aspects and more). it seems reasonable to consider supporting a symbiotic coexistence of state and local systems of tenure administration. The need for legibility making at the human scale is quickly being replaced by intelligent computer systems. Such systems are able to deal with complexities largely out of the reach of the average human expert. They interpret huge amounts of complex information and provide summarized information in a form that human experts can use and act on. We propose a deeper focus of using such systems in the domain of land administration. SmartSkeMa is a system that exemplifies how such high-tech solutions can make a difference. The role of these technologies should not be to help replace customary ways of managing access to and use of land but to make the systems of rules, norms, and ways of knowing that govern customary land tenure legible from a state perspective as is. In the guiding principles for country implementation of Fit-for-purpose land administration, it is proposed that lands with customary tenure "can be included in the formal system by demarcating the outer boundaries while retaining the community institutions that allocate and manage individual and household plots, with the option to register these land rights as the need arises" [3], (pp. 26–27). Systems such as SmartSkeMa can be used for registration of land rights within the inner boundaries of the customary land.

This paper presents the SmartSkeMa land data documentation approach built around the SmartSkeMa software system. SmartSkeMa is a portable cross-platform tool that allows people to document their land tenure rights using the familiar process of sketching or tracing and using concepts from their everyday experiences of exercising those rights. It achieves this fit by applying methods from Computer Vision and Artificial Intelligence. A sketch map is automatically digitized and (approximately) georeferenced using either numerical or logical approximation. Once the sketch is digitized and georeferenced social information can be attached to the detected objects or parcels. The social information must conform to a set of concepts actually used by the community where the tenure documentation is being conducted. The relevance of using predefined concepts is that the data can then be automatically interpreted using relations between those concepts. This is where the Artificial Intelligence methods of knowledge representation and reasoning are used to give life to cultural knowledge and information. It is possible, for example, to infer that an individual's right to use a particular piece of land under either or both the customary and legal regimes using information about the social relations in which she participates (e.g., marriage, living in a particular community, etc.)

SmartSkeMa was developed as part of the "its4land" H2020 project [4,5]. The its4land project developed four technological solutions for land tenure data acquisition. The tools were required address actual needs and readiness of their target user groups. They were also expected to be implementable and scalable within the contexts in which they were expected to be used. The guiding principles for the tools' development therefore included consideration of the seven Fit-For-Purpose (FFP) elements [6] which recommend that tools be: 1. **Flexible** enabling the capture of information about the different uses and occupations of the land; 2. **Inclusive** by covering all types of tenure and all types of land 3. **Participatory** in the manner that data are captured and used 4. **Affordable** to use for government and society at large; 5. **Reliable** for acquiring authoritative and updated information; 6. **Attainable** within a short timeframe and with the available resources; 7. **Upgradable** in response to social, legal, and economic needs and opportunities. We present an evaluation of SmartSkeMa based on an expert survey and focus groups held in Kenya. We then discuss the results in the context of the seven FFP elements as a frame of reference.

The next section gives a brief overview of the state of the art in alternative land tenure documentation tools that can be used in customary tenure regimes. We also note

the structure of conventional land administration systems through reference to the ISO 19152:2012 standard Land Administration Domain Model (LADM) [7,8]. SmartSkeMa has the ability to interpret local, i.e., customary, land concepts in the language of the LADM. This is described in more detail in Section 3 where the authors present the SmartSkeMa workflow and provide a quasi-tutorial of how SmartSkeMa works. In Section 4 we present the evaluation of SmartSkeMa. We conclude the paper with final remarks in Section 5.

## 2. Background

SmartSkeMa is a fit-for-purpose land tenure documentation approach. This section provides a context for land tenure documentation with SmartSkeMa. We give an overview of a few fit-for-purpose land registration and land tenure documentation approaches and outline how the SmartSkeMa approach differs from them.

### 2.1. Land Tenure Relations

In most official land administration institutions, the mapping of land units and the registration of the interests of them are carried out by separate functions (e.g., Departments or even Ministries). In project based, systematic land registration, on the other hand, these processes are often combined into a single stream-lined process. The distinction is between sporadic and systematic registration of land [9] (pp. 349–354).

Regardless of the 'fashion' in which land registration is executed, there are still common steps that can be generally considered to be part of a land registration process. With respect to SmartSkeMa the most important of these include (i) formalization of land tenure concepts and relations, (ii) mapping land units including parcels but also other identifiable land resources, (iii) recording of the parties involved, and (iv) documentation of the relations between parties and land units—in LA terms, rights, restrictions, and responsibilities [9] (pp. 34, 38).

In the formal land administration domain the formalization of land tenure concepts is achieved by the presence of relevant laws and statutes to govern and secure the exercise of rights on land as well as stipulate the restrictions and responsibilities on holders of land rights [9] (p. 213). A title to land is the evidence of a person's rights to the land [10] (p. 109). There are different classes of land titles by which a party may hold land rights including Freehold, Leasehold, Perpetual use rights, etc., and they vary in their interpretation from jurisdiction to jurisdiction. There are also differences in the extent to which land may be exploited and/or alienated depending on the title and jurisdiction. For example, during a field visit to Ethiopia, we learned from local communities around Bahir Dar that there are restrictions on land development in rural Ethiopia which stipulate that no permanent structures may be constructed in certain types of agricultural plots, a restriction not present in analogous scenarios in other countries like Kenya.

Underlying the different classes of land title is the form of tenure they represent. Williamson et al. [9] list a possible categorization of tenures based on observations of "processes used by a society to stabilize its access to land and resources" [9] (p. 333, Table 12.2). They make a distinction between formal and informal tenure but also list tenures including leasehold, freehold (i.e., private ownership), and customary, traditional, indigenous, and native tenures. As alluded by many authors [1,11,12] the subject of land tenure is complex. Malinowski [10] (pp. 317–320) seems to point out, based on his experiences in the Trobriands, that tenure on land is at the same time negotiated through social, cultural, religious, economic constructs as it plays a role in their construction. SmartSkeMa's domain models are designed to help bring out the "true" nature of land tenure in the customary, traditional, and informal settings. Where official land administration systems are given a generic structure using the LADM [7], customary or traditional or indigenous or informal land tenures can be approximated by SmartSkeMa's domain models.

### 2.1.1. LADM

The LADM provides a standardized global vocabulary for land administration. It specifies the general elements expected within a Land Administration System arranged in three main packages covering parties, rights, and tenure information (basic administrative units, RRRs) and spatial units. The spatial units package has a surveying and representation subpackage [7].

In the LADM land is represented as spatial units. Rights are modelled by a combination of an RRR and a basic administrative unit which is a conceptual object through which specific rights are related to spatial units. While traditionally spatial units were considered to be well defined polygonal geographic features, the advent of the fit-for-purpose paradigm required the inclusion of more flexible spatial representations of spatial units. Lemmen et al. [13], introduced the notion of levels to allow the representation of spatial units using different representational forms in the LADM. In addition to points, lines and polygons [13] also describe text-based and sketch-based spatial units. However, in its raw form, this information cannot be automatically interpreted by a land administration system: a sketch in raster format or a text in ASCII format has no spatial reference and its contents (unless properly annotated) cannot be understood and [spatially] operated on by a computer. SmartSkeMa supports the spatial representation of non-precise geographic features such as those included in sketch maps, textual descriptions, and similar data sources using qualitative spatial representations.

### 2.1.2. Customary, Traditional, Indigenous, and Native Tenures

The LADM provides a conceptual model of land tenure information. Modelling of land administration processes is outside the scope of the model. Those processes support all required transactions to keep the data up to date. The LADM concept of a Basic Administrative Unit (BAU) facilitates the abstraction the rights-to-land relation such that a customary, informal, or any other social tenure relationship also can be recorded [7] (p. 2). Through the BAU a unique or homogeneous set of rights, restrictions, or responsibilities can be associated with a group of spatial units treated as a single administrative entity. Examples of the application of this structure to a customary tenure system are provided in Annex C of [7].

Tenure, however, is much more complex and dynamic than the models represented within modern Land Administration Systems. Malinowski in his seminal chapters (Ch. XI and XII) on Land Tenure in the Trobriand Islands noted that (notwithstanding Colonization as a wholesome phenomenon in its own right) the focus on legal categories may have been the source of many land related problems in the British colonies [11]. The observation is not unique. C.K. Meek's Land Law and Custom in the Colonies [12] is full of examples of simplifications in land tenure representation that had catastrophic outcomes.

The distinction between the de facto and the de jure is that the former is practical and the latter legible [1]. The illegibility of de facto customary tenures makes them inappropriate or ill-structured for the land management functions of land administration officials [1]. However, this is not a truism. The incapacity of officialdom to grasp and control de facto organic systems of land tenure is the main reason for which official systems abstract from the multitude of rules and rule systems to a few fixed categories of tenure (hence, Seeing like a State). Although modern technology has made approximating such complex and various tenure systems more practical, one is hard pressed to find information systems and tools that support the documentation and execution of local practices related to land tenure.

### 2.1.3. STDM

Initiatives to capture customary tenure in modern LAS include the Social Tenure Domain Model or STDM [14]. STDM refers both to a tool and the land tenure domain model it implements. The STDM model is based on the continuum of rights concept which holds that land tenure is not a unitary concept but exists on a formality-informality spectrum [14,15]. STDM is related to the LADM and most of its classes have counterparts

in the LADM. This relationship is also described in Annex I of the LADM specification [7]. The idea of the spectrum, which is a simplifying model, allows different kinds of tenure to be arranged according to a fixed dimension—i.e., the extent to which tenure is enforced by state apparatuses. Rights in STDM are represented by a variety of tenure categories and it supports extension of the data model through the addition of new custom categories. For example, if special rights hold for mountain side areas then a user could add "mountainside-parcel" as a spatial unit type. However, it is not possible to infer the special rights from the types of parcels to which they apply or from other rights. That is, STDM records the static aspects of land tenure. In SmartSkeMa on the other hand, rights are transient (i.e., dynamic) [16]. A right exists contingent on other conditions which may be physical or abstract in nature, as is illustrated in Section 3.2.5 below.

*2.2. Land Tenure Mapping*

Both the collection of tenure information and the enforcement of laws to protect land rights depend on an adequately clear identification of the land involved. Many approaches have been employed for identifying land in the records and on the ground. Here we look at only a few approaches which we consider as a continuum to which the SmartSkeMa approach belongs.

2.2.1. Ground Surveys in Sporadic Land Registration

In traditional land administration the most commonly used mapping approach is the ground survey in which the positions of boundary points for a parcel or other unit of land are recorded using bearings and distances or as coordinates. Ground surveys are mostly used for sporadic land registration. The challenge here is the cost factor. Professional fees, equipment costs, and time are billed at a premium. This is impractical for systematic large scale cadastral mapping—e.g., in Cambodia a communal land titling project started in 2013 cited a 40 USD per hectare cost [17].

2.2.2. GPS and Orthophoto Maps in Systematic Land Registration

Systematic land registration requires more flexible and cost-effective approaches. One approach is to use low-cost location measuring instruments such as GPS/GNSS devices. While high-end GPS or GNSS receptor instruments are sometimes used with electronic survey instruments such as the TotalStation, for large-scale projects the low-end cheaper versions are a more practical solution. As demonstrated in the Rwanda LTR project aerial image printouts can be deployed en masse because they can be operated with little to no training reducing the human resource and hardware cost [18].

Another commonly used cadastral mapping approach involves the use of aerial imagery. The LTR projects in Rwanda and Ethiopia both used aerial (orthorectified) imagery for identification of visible boundaries. The procedure for this process is often standard. Landowners/users identify their boundaries in an image resolving differences through site visits where necessary. Boundary features marked on the aerial image are digitized in a GIS. Two approaches are prevalent. A GIS operator may look at the boundary markings on the image and mark the corresponding points in the GIS using the original correctly projected version of the image as a background guide. This usually results in many errors. The digitizer may perceive markings wrongly, he/she may have jittery hands that cause boundary points to be off from where he/she intended them, or he/she may introduce his/her own interpretations to resolve apparent ambiguities (do the hedges forming a boundary belong to the left or right side parcel?). A much more stable approach involves marking the printed orthomap with known ground control points. In this case the digitization is performed on top of a rectified version of the marked and then scanned aerial image. This means that the GIS operator simply follows the ink as marked out in the field. This somewhat reduces the human error inherent to manual digitization. The Village Land Use Planning project in Tanzania used this digitization technique for large-scale recurrent village planning exercises [19]. The advantage of drawing boundaries over orthorectified

aerial images as compared to taking boundary Coordinates on the ground using GPS is that the mappers do not need to visit every boundary corner in case of difficult terrain.

### 2.2.3. Open Tenure and SOLA

Open Tenure is an open source software application that supports mapping of land parcels or territories (demarcation), association of land users and claims to mapped parcels, and generation of summaries and reports including maps [20]. It runs on a mobile device but can be connected to the SOLA registry server which provides centralized storage and management of land tenure information [20]. The authors claim that SOLA provides functions and services provided by a typical land office. This suggests that SOLA is generic cadaster and registry system that does not take specific account of the peculiarities of different land tenure regimes. For customary tenure, flexibility, participation, and ownership are key values. These values should be part of the key design principles guiding the development of any tool that targets customary or other non-statutory land tenure regimes.

### 2.2.4. Sketching Land

It may be argued that sketching is the preferred technique of mapping for land sector non-governmental organizations (NGOs). For NGOs hand drawn sketch maps are a particularly useful tool in community mapping as they facilitate collaborative map creation among participants involved in the exercise [21,22]. Tenure mapping in these settings focusses on the community's own knowledge and consensus is achieved in a participatory manner. This is exemplified by the works of such organizations as Namati through their numerous mapping guides including the most recent Community Facilitators Guide to the Community Land Act [23] which aims to support communities in Kenya to register their community lands pursuant to the provisions of the Community Land Act (2016) of Kenya. The sketching process in the manuals focuses on the organizational aspect of the exercises: who will participate in the mapping and to what extent, how will negotiations be structured, how should the data be stored and used, etc. SmartSkeMa is a candidate tool to provide operational support to make processes such as Namati's easier to implement in a digital environment. Rugema et al. [22] also present a tool for sketch mapping but using a digital pen. By contrast SmartSkeMa is based on ink and paper sketching.

### *2.3. Summary*

The report on accuracy assessment of unconventional mapping tools published by Spatial Collective [24] indicates that simple handheld GPS devices as well as consumer smartphones with embedded GPS receivers can meet the requirements in certain conditions. Spatial Collective's study particularly showed that with little training local citizens of even remote communities were able to conduct mapping by themselves. This increases participation but still has technological and skills barriers. Besides, only one person can command a handheld device at a time. An alternative is to use sketch mapping. But when sketch maps are stored as is, as paper or raster images, they will invariably lack the contextual details required to make them interoperable with other data sources. Other approaches go further and involve the manual digitization of the sketch maps into a GIS system, enriching the data with a geographic context [20]. However, even in this case the data may fail to accurately capture significant social values and norms that govern human relationships on land. Land registration tools for customary, indigenous or other informal tenure regimes must therefore support both the capture of both the spatial and social/legal aspects of land information, and tool developers must take seriously accuracy on all dimensions of land information, not only the spatial one.

### 3. SmartSkeMa

The primary objective of the SmartSkeMa system is to provide a community focused land tenure documentation tool. It is composed of several components which come together to provide three main functions:

1.  Digitize sketch maps of land parcels, territories, and/or resources;
2.  Approximately georeference geometries in a sketch map via simple linear or quadratic regression model or using qualitative spatial relations such as left-of, inside, overlap;
3.  Capture local land tenure concepts and facts and translate them into statutory tenure terms using the LADM as the intermediating domain model.

As a software SmartSkeMa has two distinct data processing workflows with overlapping functions based on the type input map used. In the first workflow sketching is done over a georeferenced map, usually an aerial orthophoto. In the second, no background map is used during sketch; the sketch is produced freehand. The two workflows then proceed with analogous steps 1. Vectorize, 2. Align, 3. Add Land Tenure Attributes (*Data Processing in SmartSkeMa* in the third lane of Figure 1). Users can also review the data and go back to correct some data before exporting it for use with other information systems or software.

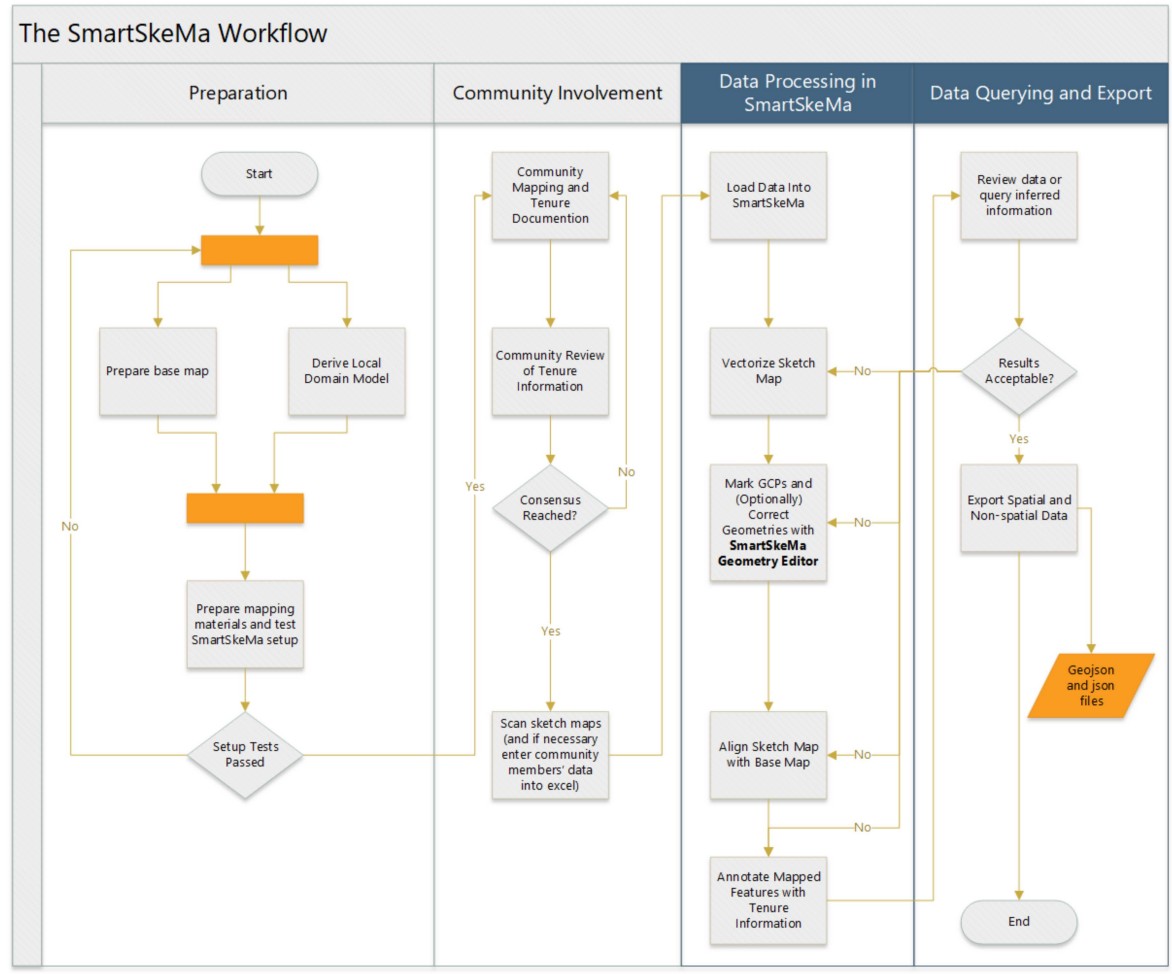

**Figure 1.** The SmartSkeMa Tenure Documentation Workflow.

Outside the software, a user must perform steps that fit into what we call the SmartSkeMa Tenure Documentation Workflow (see Figure 1). We return to the inner workings of SmartSkeMa in Section 3.2. In the next subsection we provide an overview of the SmartSkeMa Tenure Documentation Workflow.

*3.1. The SmartSkeMa Tenure Documentation Workflow*

The SmartSkeMa Tenure Documentation workflow can be integrated with other processes such as for example those promoted by Namati or other organizations. Thus, SmartSkeMa is designed to support processes that are already being applied at scale. It allows non-professionals to participate in the data digitalization process, potentially reducing time and labor cost. The key role that the steps in the current workflow play is to ensure the successful execution of each subprocess of the SmartSkeMa data processing workflow.

The first step is to prepare the system for a land data acquisition project or for a community to use on an ongoing basis (the *Preparation* lane in Figure 1). A geospatial expert is required for this step. Base reference data must be acquired and set up in the system. These can be orthophotos or vector maps that will be used for georeferencing the input sketch map data. For a given location a set of at least four ground control points (GCP) must be identified and saved to a geojson file. In addition, a domain model of the local land tenure concepts must be developed to be used during data capture and export—this can occur in parallel as seen in Figure 1.

Once everything is setup data can be collected in the community. For the sketching it is recommended to print the base map or image on an A1 or A0 size sheet (see Figure 2a). In case this is not feasible, smaller sections of the whole map can be printed on A3 sheets. Using a base map allows more precision for spatial alignment. Each printout must contain at least four visible ground control points. In addition, data sheets for recording personal and tenure information of community members must be prepared. The structure of these data sheets can be varied (there is no prescribed format) but they must be able to record personal details, relations between people and land objects, and any special conditions that apply to these relations. As we will see such conditions might be qualifying conditions or required dependencies for a relation to validly exists according local norms.

Field data collection (2nd lane Figure 1) can follow any procedure provided maps are drawn with a clear marker. In the current version sketching must be done on a transparent sheet lain on top of the base map printout (Figure 2c,e). The GCPs must be marked, preferably with a differently colored pen and pen thicknesses should be relatively similar. Field data can be scanned with a scanner for better quality results although a camera can also be used to take a photo the sketch. With digital versions of all their field data at hand (e.g., as in Figure 2d,f), the user can then proceed to produce vectorized, georeferenced and interoperable records.

*3.2. The SmartSkeMa Data Processing Workflow*

SmartSkeMa is browser-based software application that can be installed as a standalone system on a user's local computer or be accessed remotely over the internet. For purposes of portability, SmartSkeMa has been packaged using a tool called Docker (https://www.docker.com/) which allows the tool to be run in different operating systems including Windows, Linux, or MacOS. The user interface is accessed as a web page in a browser. For the technical reader, the backend components are written mostly in python with parts written in Java and the C programming language. It integrates semantic reasoning using the owlapi and the HermiT owl reasoner. The semantic reasoner allows complex social rules to be represented and reasoned over. The reasoner also uses predefined rules to interpret concepts in the local domain models as LADM concepts (see below for more details).

The user experiences the data processing workflow as six sequential steps. The following subsections describe these steps.

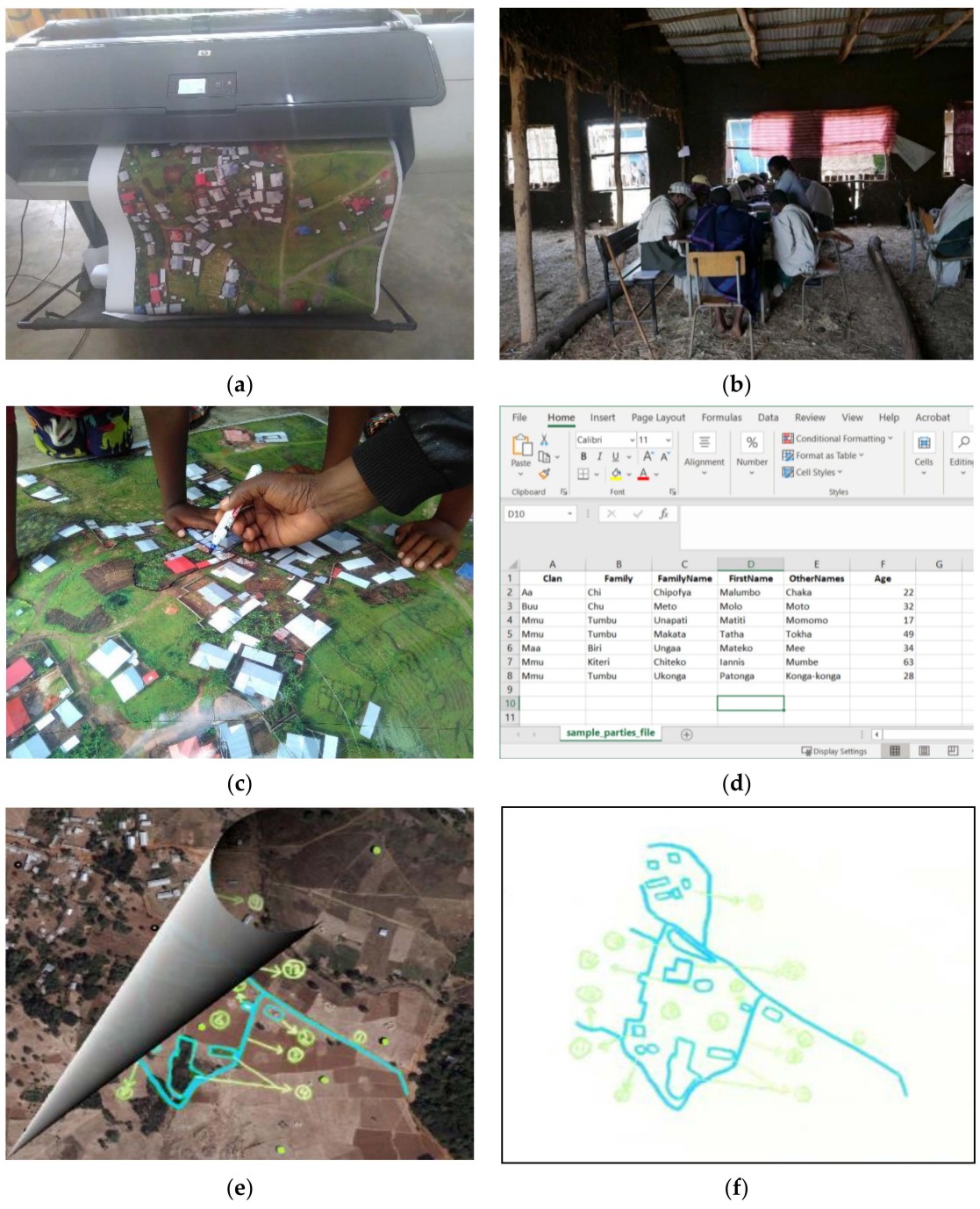

**Figure 2.** Data collection for SmartSkeMa processing. (**a**) Base map must be printed, preferably on A0 sheet. Mapping exercises can be performed in different formats e.g., in community workshop (**b**) or between small number of neighbors (**c**). Names of community members must be put into an excel sheet either directly in the field or after field work (**d**). After fieldwork the map must be scanned into a digital raster format such as *.png or *.jpeg (**e,f**).

### 3.2.1. Load Data

Once SmartSkeMa is started, the user must select the type of data processing workflow they want to execute. This is necessary for SmartSkeMa to determine the internal functions to use when processing the input data as well as the user interface elements it will need to display the results. The user is then prompted to upload all the data required depending on the selected workflow (Figure 3a). The final loaded data as they appear in SmartSkeMa can be seen in Figure 3b on the bottom.

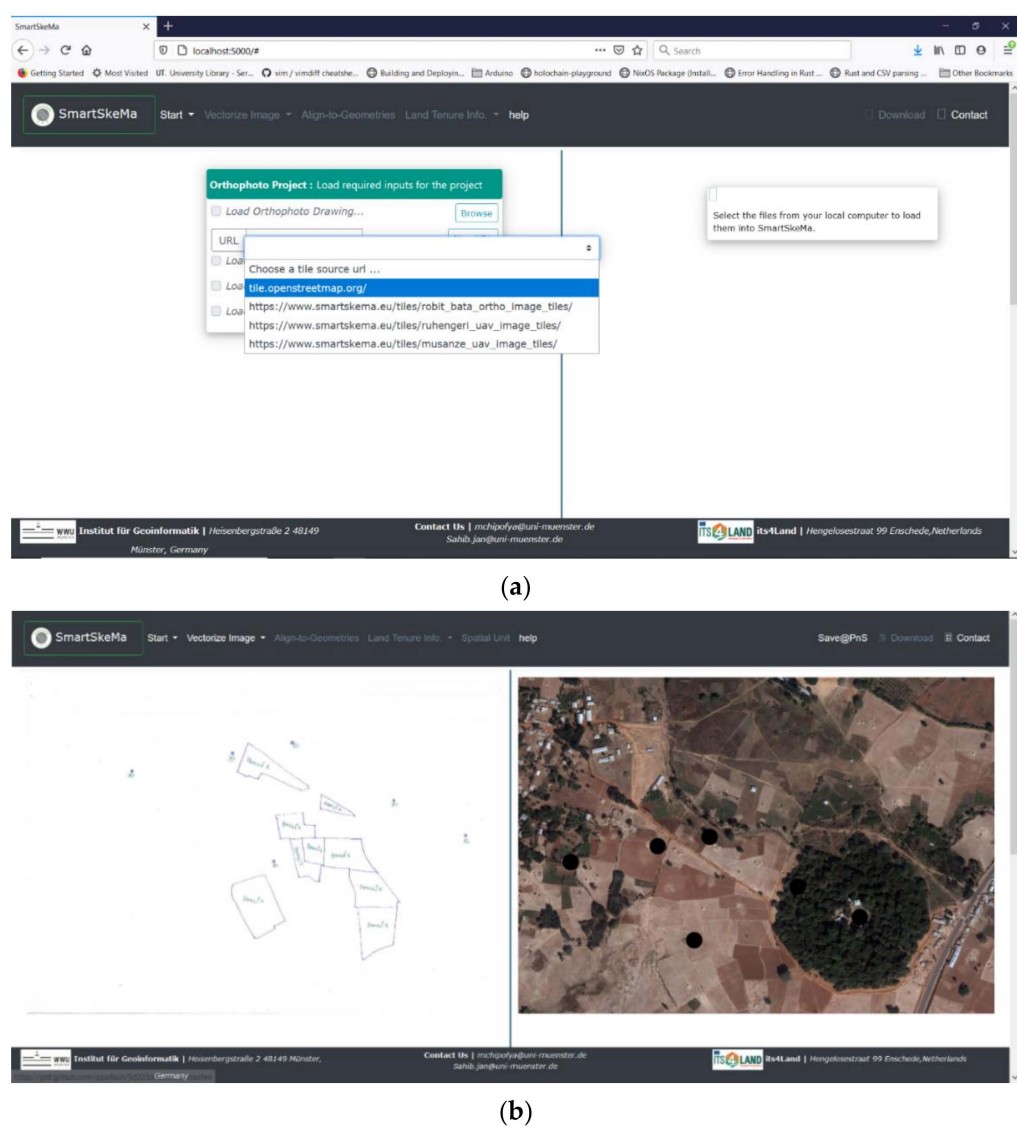

(**a**)

(**b**)

**Figure 3.** SmartSkeMa starts on a blank page. (**a**) Under the Start button users can select the type of data processing workflow they want to execute before uploading their data. In this figure, the user is specifying a source URL for the base reference data. (**b**) When spatial data are loaded in SmartSkeMa the sketch map raster image is loaded on the left panel and base map image with reference points overlayed as black dots is loaded in the panel on the right-hand-side of the window.

### 3.2.2. Vectorize the Sketch

Vectorizing the sketch map requires only the click of a button. In the background, one of two processes is invoked depending on the selected workflow. In the case of a sketch map with GPCs and no other symbols included, the system first separates image into layers of different colors. Then each layer is processed separately computing first the corner vertices of each geometry (Figure 4). From the corner vertices and the ink extracting the connections between the vertices to form topological structures called a doubly connected edge list (DCEL) [25] (pp. 29–33). The final polygons are obtained from this DCEL structure.

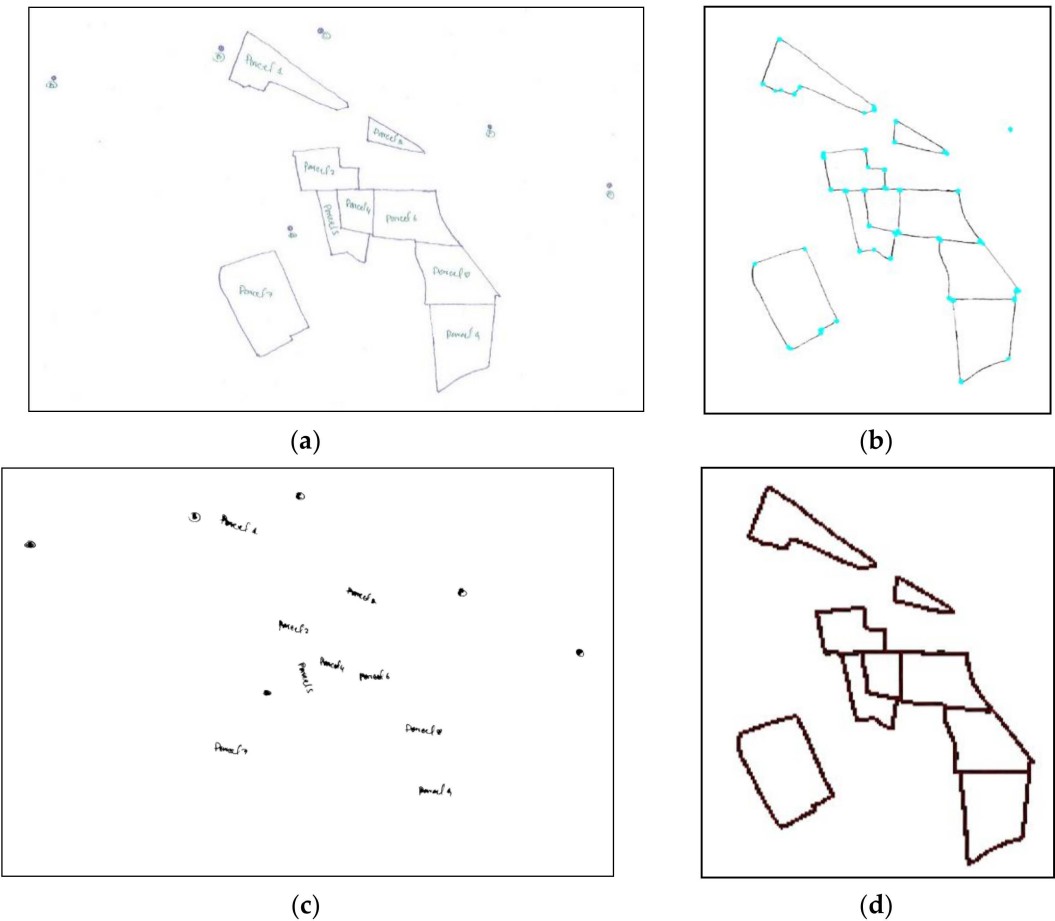

**Figure 4.** Automatic vectorization of an input vector map. (**a**) The input map, (**b**) corner vertices overlaid on polygon layer of input map, (**c**) text and markers extracted from the input map, (**d**) edges/lines of the polygons extracted shown in red.

### 3.2.3. Label Ground Control Points in Sketch Map and Edit Geometries If Necessary

Figure 5 shows the sketch map after vectorization and alignment. SmartSkeMa provides an editor function for the user to be able to adjust the geometries manually if necessary (editor buttons appear above the sketch map after vectorization is complete as seen in the left-hand image in Figure 5). The editor is also used to annotate the GCPs in the sketch map with the correct labels or identifiers so that they can be aligned correctly.

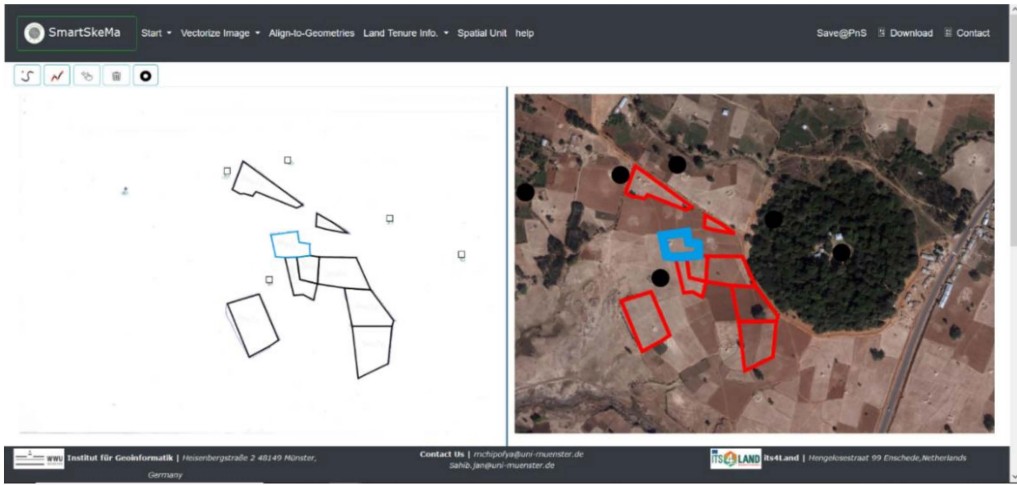

**Figure 5.** Vectorized sketch overlaid on original sketch map image and georeferenced version overlaid on base map.

### 3.2.4. Align the Sketch Map with the Base Map: Approximate Georeferencing

Georeferencing is also achieved by the user with one click of a button. The user clicks on Align-to-Geometries and in the background SmartSkeMa transforms the coordinates of the polygons in the vectorization of the sketch map into the coordinate system of the base map using the GCPs as anchor points.

Here we use a simple polynomial regression model. That means GCPs are paired: one from the sketch map to one from the base map. Each GCP has a x- and a y-coordinate. Assuming linearity in the relationship between the sketch map and base map, one forms a systems of simple linear equations relating the left-hand side coordinates to the right-hand side ones with two unknowns that become the coefficients of the transform from the coordinate space of the sketch map to the coordinate space of the base map. This is similar to defining an affine coordinate transformation between two planar geographic coordinate projection systems [26]. The map in the left-hand side panel in Figure 5 shows the georeferenced vector layer derived from the loaded sketch map.

In case a base map wasn't used during sketching and therefore no nearly affine relationship exists between the sketch map coordinates and the base map coordinates an alternative approach is taken [27]. First qualitative spatial relations are computed using the methods reported in Schwering et al. [28]. These relations include observable facts such as the fact that a feature is contained *inside or outside, near or far, adjacent or disjoint,* etc. relative to another feature. The types of relations used depend on the topological form of the feature representation (0-dimensional, 1-dimensional, or 2-dimensional). Using the set of relations between features in the sketch map and the set of relations between features in the base map together with any feature type or feature identifying information (e.g., a specific *school* identified by its name), the most likely feature correspondences between the two maps can be computed. The algorithms used in this process have been reported in Chipofya [29] and Chipofya et al. [30,31]. The output is the set of spatial relations between the elements of the two maps where the algorithm has determined some elements as spatially coinciding with each other.

### 3.2.5. Enter Land Tenure Relations (Rights, Responsibilities, Restrictions)

The last task involving data manipulation that the user must perform is to add social and tenure information associated with the mapped features if any. SmartSkeMa provides a simple form for this which allows only drag and drop interactions (Figure 6). Basic data about persons and institutions concerned are already parsed into the domain model upon upload in the first step of the workflow (Section 3.2.1). These pre-parsed data are shown at the top row in the form in Figure 6. The user selects individuals, groups of individuals, institutions (incl. clans and families) drag them to the Selected Related Elements row. S/he must then choose the tenure relations to declared (under Activities and Statuses) and drag them to the Selected Related Elements row. S/he can also drag conditional facts into the last position of the Selected Related Elements row as explained at the beginning of this section. The ⊞ button confirms the selected relation. This is repeated for as many relations as there are to be declared.

On the user side everything works as if they are simply manipulating text objects. However, the system will reject any relation that is invalid according to the local domain model defined for that particular session of SmartSkeMa. This is intended to be a domain model representing the local or cultural rules relating to land within the community being mapped.

So how does this domain model look and where does come from? The domain model is a Web Ontology Language (OWL) model. It is simply a conceptual tree structure where the nodes represent the concepts of interest together specification about binary relations between any concepts in the tree. The relationship between child and parent nodes in the tree is that the child node is a subconcept of the parent.

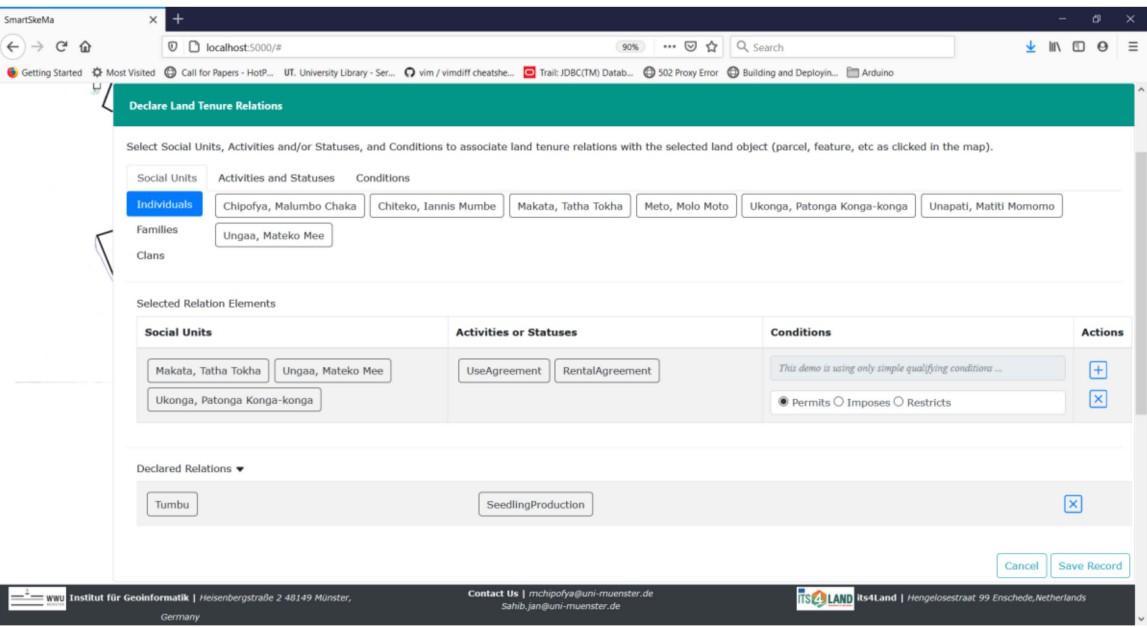

**Figure 6.** Form for declaring social relations and associations recorded during data collection in the field.

Figure 7 shows the top two levels of the concept tree of a local domain model used during one of our studies called the MSKDM [32]. The domain model has high level classes *Activity*, *SocialUnit*, *Livestock*, and *EnvironmentalCharacteristic*. As an example, Maasai culture revolves around their semi-nomadic pastoralism. Thus, a Maasai clan and its members can be represented as *SocialUnit* instances. Grazing their *Livestock* is an *Activity* which can be carried out in a geographic region that is modeled as an *EnvironmentalCharacteristic*. The significance of having geographic regions modelled as *EnvironmentalCharacteristic* is that in this particular domain the most significant aspect of tenure is not the specific geographic location but rather its ecological properties. More examples based on the MSKDM are presented in Karamesouti et al. [32] and Chipofya et al. [16].

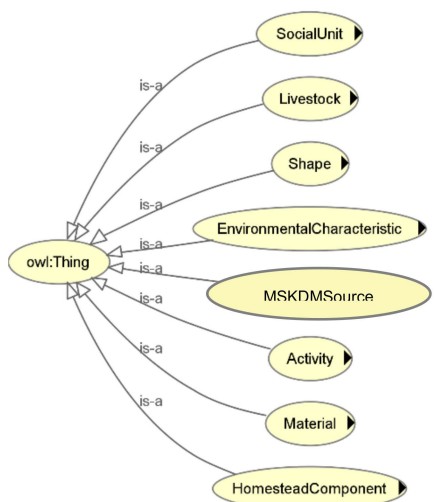

**Figure 7.** Extract of owl concept-tree from the MSKDM [32]. The top concept is called *Thing* and is a super-concept of all other concepts in the model. Each lower concept in the tree is a specialized version of its parent concept.

The reasoning process uses a series of rules and the predefined information of infer new information. Figure 8 shows a schematic of how the non-membership status of an

individual to a family can be led to the conclusion that the individual has no right to conduct a particular activity on a particular piece of land. In this case it may assumed that the parcel is the family's pasture. The rule applied behind the scenes is:

$$\text{hasValidatedExclusiveRightTo(SocialUnit: x, Activity: y)}$$

and

$$\text{isNotMemberOf(SocialUnit: z, SocialUnit: x)}$$

implies hasNoRight(z, y).

Data entered in SmartSkeMa can be translated into LADM complaint objects—i.e., objects that are instances of LADM classes. This part is also based on the reasoning capabilities described above. The idea is to interpret certain instances as Parties, other instances as RRR, others as Basic Administrative Units, Spatial Units, etc. For example, the Activity *WaterAnimalsRanch1* can be interpreted as a right with *Kashu Family* being the party. A detailed description of the design and workings the local domain model can be found in [16,32]. The rules for mapping from local domain model concepts to LADM concepts are detailed in [16].

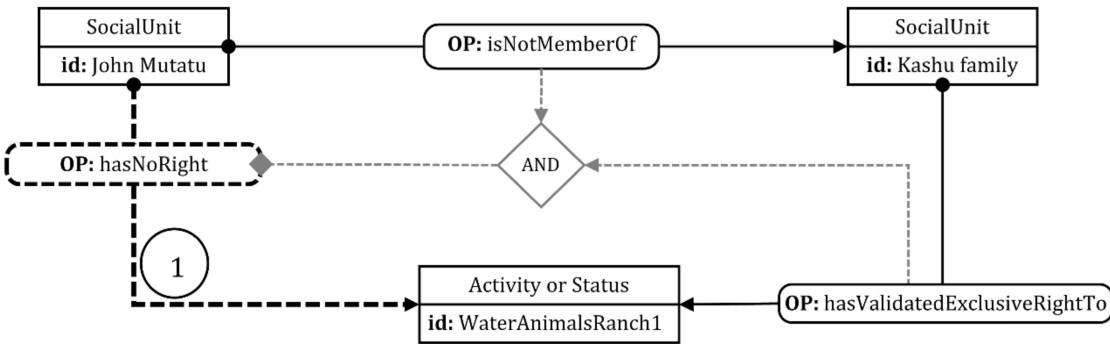

**Figure 8.** Schematic of reasoning process by which the reasoner infers new facts from input data in the domain model [32]. Rectangles represent concept instances, rounded rectangles represent relations between concept instances, and diamond shapes are logical connectors that take one or more relations as inputs and it outputs other concept relations. Here the output is the relation *hasNoRight* between the SocialUnit instance *John Mutatu* and the Activity instance *WaterAnimalsRanch1*.

### 3.2.6. Download the Data to a Chosen Location

After all processing is complete, the data can be downloaded to a folder on the local computer and be used in further processes.

### *3.3. Summary*

SmartSkeMa implements a variety of functions designed to make the digital processing of data obtained through participatory mapping in customary and informal setting more streamlined, simple, accurate. However, it is more than that. By attempting to represent local norms, rules, and recognized social relations, SmartSkeMa brings the prospect of making customary land rights mainstream.

## 4. Expert Impressions of SmartSkeMa

During the development phase of SmartSkeMa Land Administration Experts from several countries were engaged to gain an understanding of how experts judged the value proposition of SmartSkeMa. At workshops in Kajiado and Nairobi, Kenya, 20 experts answered questions regarding how and where they believed SmartSkeMa could be applied in Kenya.

In this section we present results from the workshop. In the discussion we compare the results from the study to another study where participants were asked to rank SmartSkeMa, among three other tools, on each of the seven elements of the FFP-LA paradigm [33].

*4.1. Materials and Methods*

The workshops were organized in three sessions with a roughly equal number of participants in each participating in each session. The participants were invited to the workshop to obtain a mix of land sector actors with different roles and expertise. The group consisted of:

1.  Kenyan government officials from the departments of surveys and physical planning, the land registrar's office, and the National Land Commission;
2.  Land Administration researchers from four Universities in Kenya;
3.  Private companies involved in the land sector in Kenya;
4.  Non-governmental Organizations involved in land rights mapping and advocacy.

In each session a demonstration of SmartSkeMa was presented to the participants. This included a presentation of the preparation steps, the data collection process, and a display of the tools used in the field. The demonstration was based on data collected in a field study in Kajiado, Kenya. Participants were allowed to ask questions about the SmartSkeMa Tenure Documentation workflow and the tool's functionality.

The second part of the session comprised of discussions about the applicability of SmartSkeMa for community land documentation in Kenya. The discussions were guided by a set of questions which participants were first requested to answer independently on a questionnaire before sharing their responses with the group. The discussion was then used to seek clarification on the responses shared by individual participants. The written responses consisted a Yes/No/Maybe option and free text part for verbose descriptive answers to justify the Yes/No/Maybe response.

*4.2. Results*

The results presented here focus on four questions. The verbose responses were coded by two researchers as recommended for qualitative analysis [34] and then combined to obtain the summary justifications given by each respondent. Participants were asked to indicate whether the tool could:

1.  be used in conjunction with standard land administration systems at county or national level;
2.  be used independently by community-based organizations, national civil society organizations, faith-based organizations working in the land sector;
3.  enable communities to register and govern their lands and natural resources locally according to local cultures and customs.

Pie charts summarizing the responses are shown in Figure 9. In response to question 1, 2 out of 20 respondents thought SmartSkeMa could not be used in that context, 4 indicated that there may be some use for it but also raised doubts (response: May be), while 14 responded with a 'Yes'. The justification given for the 'No' responses were related to low expected accuracy of the mapped points. The most common justifications for the 'Yes' responses were (i) interpretation of the actual relations on the ground, (ii) mainstreaming or capturing the community perspective, (iii) the participatory nature of the tool, and (iv) the ability to map shared or communal resources. These correspond well to the response to the next question presented below.

For question 2 the one respondent who responded 'No' gave the reason that "Ground truthing is necessary and the basemap [must be] well done to show what one has collected in the field—especially on legal [aspects]". This is of course anti-thetic to the assumptions behind SmartSkeMa's design. 17 participants responded with 'Yes' indicating conflict resolution, planning, capturing attribute and legal data as potential applications. 'Maybe' respondents expressed that its application would be require professional guidance.

The reasons given for the 'Yes' responses to question 3 were quite varied but the dominant themes included

1.  potential to reduce conflicts;
2.  use in resource use/management and planning (also at the community level);

3.    And related to the above, that it is useful for planning purposes.

Unsurprisingly, those who answered 'No' mentioned that there were already preexisting legal restrictions on permissible land uses and that the outputs would not meet legal thresholds for surveys.

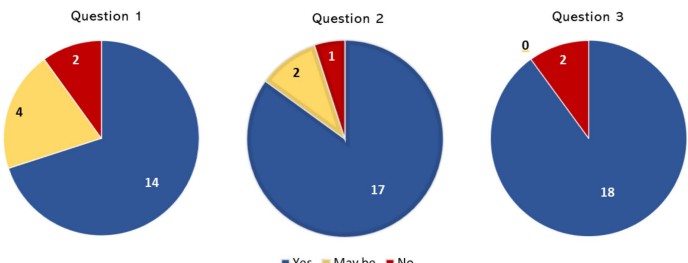

**Figure 9.** Expert opinions on whether or not SmartSkeMa could be used (i) in conjunction with standard Land Administration systems, (ii) by non-governmental land sector organizations, and (iii) to enable land tenure self-governance by communities.

Respondents were also asked to mention areas where they saw SmartSkeMa being applied within their domain of work. The responses were summarized into categories and the most common categories are shown in Figure 10. The most prevalent application areas fell under the category of "Participatory capture of community view on land (mapping/appraisal/consultation)". However, respondents also mentioned application areas which we grouped under the following categories with high frequency (7): conflict resolution, land use planning, and documentation of varied tenure information.

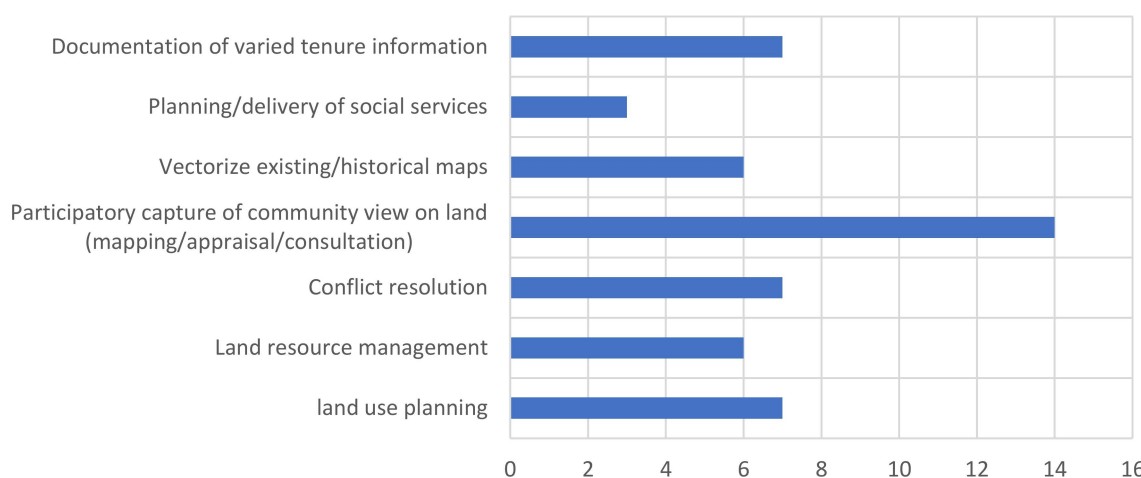

**Figure 10.** Expert opinions on the possibility of using SmartSkeMa in conjunction with standard land administration systems.

### 4.3. Discussion

The observations made by respondent in the study reported above were matched with a separate survey [33] conducted in January 2021 where respondents were asked to rank SmartSkeMa on each of the Fit-For-Purpose dimensions of Flexibility, Inclusivity, Participation, Affordability, Reliability, Attainability, and Upgradability.

Of the 14 participants who took part in the latter survey 10 rated SmartSkeMa with a score of at least 4 on a 5-point scale for the dimensions Flexibility, Inclusivity, and Participation while 7 participants rated the Reliability dimension with a score of 3 (don't know) indicating, among other things, that data produced with the tool could not be used

as "authoritative data" and would not meet "legal precision and accuracy requirements"—see [33] for more details. This result should not be surprising. Rahmatizadeh et al. [35] in a study on parameters that affect how fit-for-purpose spatial data collection tools are selected also found that over 50% of respondents considered accuracy to be one of the two most important criteria.

The results above are significant as they highlight what the expert community considers as the weak points for the SmartSkeMa system and its workflows: it has not yet been proven to meet legally stipulated accuracy thresholds for geolocation (i.e., spatial) measurement in any jurisdiction. This is significant to attain government buy-in and acceptance of data collected with the tool. Experiments by Asiama et al. [36] in the Dagbon region of Ghana found that although the accuracy of mapping with a GPS equipped mobile phone ranged between 1 m and 5 m there were not many disputes between neighbors regarding the position of mapped parcel borders. This could be explained by the fact that during the mobile mapping all neighbors were always present [36].

Buy-in can also be achieved through acceptance by the community as a whole including local authorities such as chiefs and community/village heads men and women. The Global Land Tool Network and its partners, for example, achieved a community level of acceptance in their land rights documentation projects in Zambia [37] and Uganda [38] through engagement of the chiefs and the community. The projects used STDM as the software tool to support mapping and attribute capture.

The strength of SmartSkeMa as shown in both studies is in its participatory nature and its ability to capture culturally relevant local land tenure categories. It is our impression that all other positive ratings obtained from this study directly follow from these two attributes of the tool. Juxtaposing the criticism of poor accuracy with the positive evaluation for participation one notices a discrepancy between the legal requirements for recognition of land records and the needs of communities whose tenure on land is not legally recognized.

On one hand mapping the vast areas covered by customary land tenure regimes using conventional methods within any reasonable time frame is a near impossible task. Simplifications are possible by engaging teams of non-professional mappers as done in Rwanda. But this solves only one of the problems. The solution must also account for local custom. As such legal requirements for the recognition of customary land right must accept lower thresholds for the measurement accuracy of the maps used. Conflicts, which are what most land professionals fear would result from low accuracy maps, can be dealt with using existing conflict resolution mechanisms within community structures.

## 5. Conclusions

There are several tools for documenting customary land rights. Most of these tools are designed to be easy and cheap to use. Most are also designed to work with a variety of land rights. The challenge, however, is to be able to document land rights and their dynamics as they truly function. In customary settings this is an essential gap as recorded land rights may differ significantly with the actual experiences of land users. SmartSkeMa attempts to close the gap by modeling land tenure concepts as closely as possible. This allows SmartSkeMa to support both the legibility of customary land tenure to government authorities and the preservation of the customs within which the tenure relations operate. Preserving customary rights to land requires also preserving customary ways of allotting, negotiating, and exercising those rights. Otherwise, the entire notion of customary land tenure itself becomes a shell or a cover for replacing customary tenure with statutory tenure.

In this paper we introduced the functionalities of SmartSkeMa, presented a workflow for using it, and described some the underlying technical details. SmartSkeMa has functions to make participatory mapping and recordation of land rights accessible to non-professionals. Setting it up, however, may require a good understanding of several diverse skills. An OWL ontology needs to be set up, base map data need to be obtained, and field work materials need to be prepared. Once the setup is done, however, only a little training may be required to allow a community to independently document their lands.

Overall SmartSkeMa received positive feedback from two separate groups of experts in two different studies. SmartSkeMa's strength is in its ability to include detailed local knowledge into a land information database. Many respondents to our first study noted that SmartSkeMa had the potential reduce land related conflicts. This is a point to be investigated further in actual use of the tool in practice.

Going forward there are two directions for advancing our work. First, we will eliminate the need for using a transparent overlay in order to trace over a map. One approach to do so is to apply image registration on the scanned field sketch with its original. If good approximate alignment can be achieved, then taking the image differences would eliminate original patterns leaving only the sketched ink. The other extension has to do with facilitating communities' self-governance of land tenure. We believe this could facilitated by user friendly tools built on top peer-to-peer overlay networks and decentralized systems, including blockchain networks.

**Author Contributions:** Conceptualization, M.C.C., S.J. and A.S.; methodology, M.C.C. and S.J.; software, M.C.C. and S.J.; formal analysis, M.C.C.; data curation, M.C.C.; writing—original draft preparation, M.C.C.; writing—review and editing, M.C.C. and S.J.; visualization, M.C.C. and S.J.; supervision, A.S.; project administration, A.S. and M.C.C.; funding acquisition, A.S. and M.C.C. All authors have read and agreed to the published version of the manuscript.

**Funding:** Funding of the publication costs for this article has kindly been provided by the School of Land Administration Studies, University of Twente, in combination with Kadaster International, the Netherlands. The research was partially funded by the research project "its4land," which is part of the Horizon 2020 program of the European Union, project number 687828.

**Institutional Review Board Statement:** Not applicable.

**Informed Consent Statement:** Not applicable.

**Data Availability Statement:** Not applicable.

**Acknowledgments:** The authors acknowledge the contributions by other members of the its4land project to the successful completion of the fieldworks and workshops.

**Conflicts of Interest:** The authors declare no conflict of interest. The funders had no role in the design of the study; in the collection, analyses, or interpretation of data; in the writing of the manuscript, or in the decision to publish the results.

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
