# Peer review of "SmartSkeMa: Scalable Documentation for Community and Customary Land Tenure"

_land, doi:10.3390/land10070662_

Round 1

Reviewer 1 Report

This paper introduces a tool to document customary and community land rights. The paper aims to an  evaluation of SmartSkeMa based on an expert survey and focus groups held in Kenya. The results of this exercise are discussed - in the context of the seven FFP elements as a frame of reference.

The methodology behind this evaluation deserves better and more explicit explanations. Who are these experts? What is their background? How were they selected? Have they worked with the tool? This needs more attention, for example in a separate section "materials and methods". This may also include the use of AI, owl, etc aspects. And also “construction. SmartSkeMa’s domain models are designed to help bring out the “true” nature of land tenure”. This could be better linked to the Fit-For-Purpose Land Administration: Guiding Principles For Country Implementation”, page 27. Here it is written:  “Also, communal lands (with customary tenure) can be included in the formal system by demarcating the outer boundaries while retaining the community institutions that allocate and manage individual and household plots, with the option to register these land rights as the need arises (Byamugisha, 2013).”. This may be an interesting bridge to the need for the tool/approach as presented in the paper. In general there can be more reference to the Guiding Principles.

A further general remark is that there could be more attention to the concepts and innovations in the paper. For example “The role of these technologies should not be to help replace customary ways of managing access to and use of land but to make the systems of rules, norms, and ways of knowing that govern customary land tenure legible from a state perspective as is.”. It could be better illustrated how this is done. Same for: “SmartSkeMa has the ability to interpret local, i.e. customary, land concepts in the language of LADM.” This disserves more detailed illustration.

The paper mentions the LADM - but without an explicit reference to the standard: ISO/TC 211 Geographic Information/Geomatics. ISO 19152:2012 Geographic Information—Land Administration Domain Model (LADM); International Organization for Standardization: Geneva, Switzerland, 2012. Reading up on this standard is recommended.

The term "customary right" is mentioned many times in the ISO 19152 standard. LADM is “spatial unit” based – not “parcel based” as assumed in this paper. See for example line 15 in the paper (“continue to assume a parcel based model”) and more specific in line 138: “In LADM land is represented as parcels or legal spaces and parties enjoy rights, restrictions and responsibilities (RRRs for short) incumbent on these well-defined parcels or legal spaces. Rights are modelled by a combination of an RRR and a basic administrative unit which is a conceptual object through which specific rights are related to spatial units.” Please notice that this explicitly not valid for LADM and STDM. Those models are not parcel based. They are spatial unit based. Parcel definitions are included in legislations and/or regulations. Based on those definitions a parcel can not concern representations informal or customary area’s. That’s why it is called spatial units. A parcel can be a spatial unit type.

The LADM standard defines a basic administrative unit as an: “administrative entity, subject to registration (by law), or recordation [by informal right, or customary right, or another social tenure relationship], consisting of zero or more spatial units against which (one or more) unique and homogeneous rights [e.g. ownership right or land use right], responsibilities or restrictions are associated to the whole entity, as included in a land administration system.”. This definition should be recognised in the paper. That will be doable – because the paper says that “SmartSkeMa has the ability to interpret local, i.e. customary, land concepts in the language of LADM”, so a more realistic alignment between the contents of this paper and the LADM (and STDM, which is included in the informative Annex I of the ISO 19152) is possible. This will not impact the presented innovations in the paper – because those innovations are process related and not so much data related.

And: line 154: “ LADM captures land tenure from an official, bureaucratic perspective”. This does not need to be true. Please rephrase.

The title of the paper is: “SmartSkeMa: Scalable Documentation for 2 Community and Customary Land Tenure”. The “Scalability” aspect needs to be related to more evidence from projects or field tests. See for example lines 73 – 76.

Please notice that keywords are missing.

There could be reference to the penmap approach when introducing sketch maps: “Rugema, D.M., Verplanke, J.J. and Lemmen, C.H.J. (2015) Digital pen method. In: Advances in responsible land administration / edited by J.A. Zevenbergen, W.T. de Vries and R.M. Bennett. Boca Raton: CRC Press, 2015. ISBN 978-1-4987-1961-2. pp. 131-144.”

Line 106: “ i) formalization of land tenure concepts and relations”. This formalization does not depend on the tool? It is a separate process? There can be “support” in formalisation with the presented tool??

Line 132: “Where official land administration systems are given a generic structure using the ISO 19152:2012 Land Administration Domain Model (LADM) [6], customary or traditional or indigenous or informal land tenures can be approximated by SmartSkeMa’s domain models.” The reviewer has the opinion that LADM covers customary tenure. In case there is need for tool related domain model there should be arguments for this.

Line 134: “The Land Administration Domain Model (LADM) attempts to provide a standardized global 134 vocabulary for land administration. “. “Attempts”?  ISO 19152 is an International Standard. Please replace by “provides”

Line 163: “The illegibility of de facto customary tenures makes them inappropriate or ill-structured for the land management functions of Land Administration officials.” LADM claims to cover customary tenure.

Line 150: “None of the existing LADM spatial profiles currently handle qualitative descriptions of spatial objects in a way that allows spatial operations on them. SmartSkeMa supports the spatial  representation of non-precise geographic features such as those included in sketch maps, textual  descriptions, and similar data sources using qualitative spatial representations”. Please add references to the studies spatial profiles or reformulate. LADM supports text and point based spatial units. Or sketches.

Line 178: “type. However it is not be possible to infer the special rights 178 from the types of parcels they apply or from other rights”/ This needs better explanation.

Line 184: “continuum” – there could be reference to: Lemmen, C.H.J., Augustinus, C., du Plessis, J., Laarakker, P., de Zeeuw, K., Saers, P. and Molendijk, M. (2015) The operationalisation of the Continuum of Land Rights at country level. In: Linking land tenure and use for shared prosperity, proceedings of the annual World Bank conference on land and poverty, 23-27 March 2015, Washington DC, United States. 27 p.

Line 201: “general boundaries”. Better to use visible boundaries as in the guiding principles (p27).

Line 243: “While tools such as Open Tenure and STDM may fall short of meeting legal thresholds for land  survey accuracy, ….”.  STDM or Open Tenure to not set accuracies….

Figure 1: the “Align” visualisation in the lower workflow is unclear. And: is “Admin source” a reference to LADM?

Line 288: “Finally, a domain model of the local land tenure concepts must be developed to be used”. Please give some brief explanation here (from Karamesouti et al. [21] and Chipofya et al. [22]). This is really interesting.

Line 311 – 316…… You aks a lot from the reader. This is difficult for most readers. Can be eliminated and explained in other words.

Figure 2. Refer to the 6 pictures as a., b., c., d., e., f. and explain accordingly in the caption. Same for similar figures.

Line 328: “results. The user is then prompted to upload all the data required depending on the selected workflow (Figure 3). The final loaded data as they appear in SmartSkeMa can be seen in Figure 5.”. Figure 5 should be figure 4? Swap?

Figure 4. Contains black rectangle? Refer to the 4 pictures as a., b., c., d. Please replace the caption by a better understandable version. Also figure 5!

Lines 364 – 371. Please explain in a more simple way.

Figure 6. Is there a link between the two pictures?? Please explain.

Figure 8. Difficult to understand. Please give a better explanation – this is interesting.

Also Figure 9. Explain the symbols?

Section 4.1. Question 2 is not discussed…

Line 492: “legally stipulated accuracy”. The experts were only surveyors? Is the “legally stipulated accuracy the reason for incomplete land administration? Please reflect on this. Same in line 505 (low accuracy maps).

Quality of English:

  • Line 43: “. Ultimately state systems must then (at least for a time) co-exist, compete with, or be rejected by and thus come into direct conflict with the existing norms.” Rephrase please. This is difficult to understand…
  • Line 67: “automatically interpret” should be “interpreted”?
  • Line 76: “and scalable within the contexts in which were expected to be used” should be “and scalable within the contexts in which they were expected to be used”?
  • Line 90: “LADM”. Should be “the LADM”. Please check the complete paper on this.
    Line 125: “As alluded by many an author”. Should be “As alluded by many authors”?
  • Line 135: “an LAS” should be “a LAS”
  • Line 178: “type. However it is not be possible”. Should be “type. However it is not possible”?
  • Line 314: “The semantic 314 reasoning is what allows concepts in the local domain models to be interpret as LADM concepts.”. Please rephrase.
  • Line 321: typo “exrcises”
  • Line 379. “bout”. ?
  • Line 388: “But they will be met with an unpleasant if they select and confirm a relation that is invalid according to the local domain model defined for that particular session of SmartSkeMa.” Something is missing after “unpleasant”?
  • Line 397: “Figure 8. Extract of owl concept-tree from the MSKDM.Figure 8 shows part of the concept tree of a local domain model used during one of our studies called the MSKDM [21].” Please rephrase.
  • Line 530: “advancing the our work”. Should be “advancing our work”.
  • Line 532: “god” should be “good”

Author Response

Dear Reviewer,

Thank you very much for your detailed review and insightful comments. We took every comment very seriously and tried to address each comment to the best extent possible. We hope that the resulting changes have improved the manuscript to a publishable state.

Please see the attachment for detailed responses to the comments.

Warm regards,

Reviewer 2 Report

The authors describe a tool that could be a game changer in fit for purpose land administration. However, the manuscript should be revised with consideration for the following comments to make it adequate for publication.

Though seemingly irrelevant, the authors’ separation of LADM and STDM into two separate subsections (2.1.1 and 2.1.3) without acknowledgement of the relationship between the two, make the two seem unrelated. The close relationship between the two concepts should be explicitly acknowledged.

The authors detail the steps of the process used in mapping using SmartSkeMa. However, a reader will benefit from a flowchart that will outline the steps which cover several pages in the manuscript.

The method of assessing the tool does not cover the seven FFP elements the authors tout as guiding principles for the development of the tool, namely flexible, inclusive, participatory, affordable, reliable, attainable, and upgradable.

Given the assessment of the tool was done by experts, one would have expected there to be more details to their assessment, rather than what seems like structured questions with little detail. With a small pool of experts doing the assessment, there should be provided more details as to their assessment of the tool.

The discussion of the results in the manuscript does not relate the study to other studies relating to fit for purpose land administration approaches, but is rather a further analysis of the results, without the placement of the results from SmartSkeMa in the wider body of literature.

Author Response

(The authors gave the same response as above.)

Round 2

Reviewer 1 Report

Thanks for the new version of your paper.

Please find some minor errors:

  • Line 357: “described” should be “describe”
  • Line 414: “… are paired one….” should”…are paired: one…”
  • Line 415:  “has an x-, and a y-coordinate”. “an” should be “a”
  • Line 427: “dimension” should be “dimensional”? 3x
  • Line 429: “base” should be “base map”?
  • Line 553: “The group consisted of” should be “The group consisted of:”?

Author Response

Dear Reviewer,

Thank you very much for your comments and observations. We have made all the specific corrections as you have indicated in your comments.

Best regards,

The authors

Reviewer 2 Report

The comments have all been addressed adequately in the revised version of the paper.

Author Response

Dear Reviewer,

Thank you very much for your favorable review of our revised submission.

Regards,

The authors